# Physical Modeling of the Scour Volume Upstream of a Slit Weir Using Uniform and Non-Uniform Mobile Beds

Ruaa Khalid Hamdan [1,2,*], Aqeel Al-Adili [1] and Thamer Ahmed Mohammed [3]

1   Department of Civil Engineering, University of Technology, Baghdad 1066, Iraq
2   Ministry of Water Resources, Baghdad 10064, Iraq
3   Department of Water Resources Engineering, Collage of Engineering, University of Baghdad, Baghdad 10071, Iraq
*   Correspondence: bce.19.55@grad.uotechnology.edu.iq

**Abstract:** This study presents new laboratory data on the time-varying scour upstream of a slit weir used for sediment release near hydropower intake. The governing parameters in temporal and spatial scour hole development under steady and unsteady flow conditions were experimentally investigated. This study includes 40 scenarios for steady flow at center and side slit weirs for uniform and non-uniform sediment with median sizes of $d_{50}$ = 0.24, 0.55 mm and four scenarios of unsteady flow conditions at a center slit weir under different flow intensities. The steel slit weirs were built in a rectangular brick and concrete flume with dimensions of 1.25 m wide, 8.0 m long and 1.0 m deep. The dimensional analysis supports recent studies. This study demonstrates an increment in the resulted scour volume for fine and uniform sands at the center slit weir of about 2 times the value of coarse sand and 1.25 times the value measured with the side slit weir for uniform and non-uniform sands. However, the resulted scour volume for fine non-uniform sand at the center slit weir was recorded as 2.5 times that of coarse sand. There was a dramatic increase in the scour volume of about 4 times at the center slit weir and 3–5 at the side slit weir when the flow rate increased by 4 times.

**Keywords:** sedimentation; hydraulic power intake; slit weir; scour volume; physical model

## 1. Introduction

The main source of sediment accumulation in water bodies (rivers and reservoirs) is fine particles conveyed by flowing streams [1]. After the identification of sediment accumulation problems upstream of dams, it gained sharp focus worldwide [2,3]. Frequent sediment accumulation in hydropower dams causes reductions in water storage, blockage of turbine intake and the disruption of power generation. A reduction in sediment deposition by sustainable methods is required. Many techniques for reducing sedimentation in reservoirs have been adopted worldwide. However, from economic perspective, the release of sediment with water from dam gates or slit weirs is a suitable method and should be studied. The method is based on generating a scour in sediment accumulation, which activates sediment releases near the hydropower intake by creating 3D tornado eddies that minimize the possibility of turbine intake clogging. The method is considered to be a potential solution that can help to avoid the interruption of power generation. It is crucial to study how to balance between volumes of water and sediment release from hydropower reservoirs in order to minimize the operational problems of both the reservoir and the hydropower plant. In the present study, experiments were conducted to study the impact of uniformity, the coarseness of mobile beds, the location of slit weirs and flow conditions on the size of the scour hole formed upstream of slit weirs. The removal of sediment upstream and downstream of weirs and other hydraulic structures have been studied by many researchers, and the studies can be categorized as experimental, numerical and statistical. Based on the published literature, it has been found that experimental studies

on the scour upstream of slit weirs are limited. Table 1 summarizes the published studies on the problem.

**Table 1.** Studies on scour upstream, downstream and around hydraulic structures and sediment releases.

| Author | Nature of Study | Main Findings |
|---|---|---|
| **Scour development upstream and around slit weirs** | | |
| Ota and Sato [4] | Sediment releasing through a dam gate | Simulated the scour process around a slit weir experimentally and numerically by a 3D numerical analysis based on Reynolds-averaged Navier–Stokes (RANS) equations coupled with the VOF method and the k-co SST turbulence closure model. |
| Ota [5] | 3D numerical model for a scour around a slit weir | Updated the study of Ota and Sato (2015) to reproduce the resulting scour around a slit weir. |
| Ota [6] | Ordinary differential equation model for the scour upstream of a slit weir | Investigation of time varying scour volume and maximum scour depth generating upstream of a slit weir under steady and unsteady conditions by adopting an ordinary differential-equation-based model. |
| Ota [7] | 3D simulation for the scour upstream of a slit weir | Suggested 3D hybrid Euler–Lagrange model for a bed-material load considering transitions between the bed load and suspended load to accurately reproduce the scour around the slit weir. |
| Nkad [8] | Scour volume upstream of a slit weir | The scour volume and maximum scour depth were investigated experimentally on the upstream side of a slit weir under steady flow, clear-water scour conditions and non-uniform sedimentation. |
| **Scour development upstream and downstream of a submerged weir** | | |
| Guan [9,10] | Scour investigation upstream and downstream of a submerged weir | The scour was investigated experimentally upstream and downstream of submerged weirs within live-bed scour conditions. New equations, including the effects of sediment size, flow intensity and weir geometry are proposed for the prediction of equilibrium scour depths, and a new design method is given for estimating the maximum scour depths at the weir. |
| Wang [11,12] | Local scour at the submerged weir | Experimentally studied the effect of different slopes downstream and upstream of a submerged weir on the scour within numerous scenarios for fine and coarse sediments under clear and live-bed scour conditions. The study presents a new technique for investigating the maximum scour depth and the correlation between average and maximum scour depth. |
| **Local scour at bed sill** | | |
| Gaudio [13] | Local scour downstream of a bed sill | Experimentally studied the influence of morphology on the scour downstream of a bed sill within a gravel bed with a classical dimensional analysis. The study presents numerical formulas for estimating scour depth, scour hole length and the location of the maximum scour depth. |
| Marion [14] | Local scour at the bed still in high-gradient streams | Experimentally predicted the effect of steady releases of sedimentation under clear water conditions on scour depth and shape, created at the toe of the bed sills. |
| **Scour around bridge piers** | | |
| Hager [15] | Horseshoe vortex of sediment-embedded bridge piers | Experimentally investigated the flow features around a circular bridge pier. The study presents novel data for numerical simulations. |
| Najafzadeh [16] | Local scour around a vertical pier in cohesive soils | An experimental work was carried out to predict the maximum scour depth generated around bridge piers under various governing parameters. The study presents a general scour depth equation and compares it with an empirical scour depth equation, and both are in good agreement. |
| Ghodsi [17] | The geometric effect of complex bridge piers on the maximum scour depth | Eighty-two laboratory tests within six physical models were adopted to study pier geometry as the affecting parameter the on maximum scour depth. A dimensional analysis was carried out, and the study results clarify that each individual parameter impacts the maximum scour depth. |

**Table 1.** *Cont.*

| Author | Nature of Study | Main Findings |
|---|---|---|
| Amini, Magdi and Truce [18–20] | Bridge scour | The majority of published studies are focused on bridge scour. |
| **Scour around different weir types and sediment-release techniques** | | |
| Dey and Barbhuiya [21,22] | Flow field in scour hole at a vertical and wing wall abutment | An experimental study was conducted to investigate the local scour and 3D flow parameters in a vertical and wing wall abutment within a clear water scour. |
| Abdollahpour [23] | Erosion and sedimentation downstream of a W-weir | Experimentally studied the effect of a W-weir structure on the erosion and sedimentation of a sinusoidal channel. |
| Liu [24] | Piano key weir | The PKW performance was evaluated and analyzed with new formulas for efficient discharge release. |
| Khalili and Honar [25] | Simi-circular labyrinth side weir | An experimental study was conducted, evaluating a semi-circular labyrinth side weir to investigate the effect of the structure geometry on the flow intensity coefficients. |
| Powell and Khan [26] | Scour upstream of a circular orifice | Investigated the sediment transport mechanism and the scour area, depth and shape upstream of a circular orifice. The investigation was conducted under steady flow conditions with different sediment sizes and heads on the orifice. |
| Lauchlan [27] | Sediment transportation over weirs | Sediment transport was experimentally predicted with steep-slope weirs and dikes, including both the bed load and the suspended load. |
| Zhang [28] | Local scour around submarine pipelines | An experimental work is proposed with empirical equations for accurate live bed scour predictions around submarine pipelines. |
| Fathi-Moghadam [29] | Desilting of non-cohesive sediment | An experimental work is presented with numerical equations used to predict the scour cone depth and volume generated throughout the flushing process from dam intake. |

## 2. Materials and Methods

*Dimensional Analysis*

Many of the experimental studies on the scour around hydraulic structures adopt the Buckingham $\pi$ theory to determine the governing parameters controlling the scour phenomena [11,12]. However, refs. [9,10] tested the hydraulic structure geometry, fluid and sediment properties, flow characteristics for estimating dimensionless relations with the scour depth, and the flow characteristics, physical soil properties and the structure geometry were tested as a controlling boundary [30]. The authors of [31] included the scour volume as an independent parameter in their study to predict the scour depth downstream of the W-weir on the sinusoidal mid bend. Dimensional analysis has been used in recent studies in order to group the most effective variables that govern the scour volume for efficient sediment removal upstream of a slit weir. The governing variables are scour volume ($V_s$), scour depth ($d_s$), slit weir height ($h_s$), weir height ($h_w$), median grain size ($d_{50}$), the density of the water ($\rho$), the density of the sediment ($\rho_s$), dynamic viscosity ($\mu$), acceleration due to gravity ($g$), the approach flow velocity ($v$), armor velocity ($v_a$), sediment entrainment critical velocity ($v_c$), flume width ($B$), slit weir width ($b_s$) and flow water depth ($y$). The independent variable and dependent variables can be described by the following formula:

$$V_s = f(d_s, h_s, h_w, d_{50}, \rho, \rho_s, \mu, g, v, v_a, v_c, B, b_s, y) \tag{1}$$

The independent parameters are the flow parameters ($v$, $y$), flume and slit weir geometries ($B$, $b_s$, $h_w$, $h_s$), water characteristics ($\rho$, $\mu$, $g$) and sediment characteristics ($\rho_s$, $d_{50}$, $v_c$, $v_a$), and the dependent parameter is the scour volume $V_s$.

After conducting the dimensional analysis, the governing dimensionless parameters can be written as:

$$\frac{(V_s)^{\frac{1}{3}}}{d_s} = f\left(F_r, \ R_e, \ \frac{\rho_s}{\rho}, \frac{d_{50}}{d_s}, \ \frac{v_a}{v}, \ \frac{v_c}{v}, \ \frac{y}{d_s}, \ \frac{h_w}{d_s}, \frac{h_s}{d_s}, \frac{b_s}{d_s}, \ \frac{B}{d_s}\right) \tag{2}$$

The following parameters have no effect on the studied problem ($\frac{h_s}{d_s}$, $\frac{h_w}{d_s}$, $\frac{b_s}{d_s}$ and $\frac{B}{d_s}$) since it was not changed during the tests. In addition, the water and sediment densities were considered to be a constant. However, the Reynolds number is dominant in close conduit flow [30]. Therefore, the Froude number is the dominant parameter in this study, as it is adopted in many different hydraulic structures such as weirs, spillways, stilling basins, etc., under free surface flow conditions. Furthermore, the sediment gain size, sediment armor, entrainment velocity and the flow depth change with different flow intensities. Therefore, the dimensional analysis can be written as:

$$\frac{(V_s)^{\frac{1}{3}}}{d_s} = f\left(F_r, \ \frac{v_a}{v}, \ \frac{v_c}{v} \ , \frac{d_{50}}{d_s}, \frac{y}{d_s}\right) \tag{3}$$

The experimental work results were implemented into the present dimensional analysis. Figures 1–4 show the relationship between $\frac{V_s^{\frac{1}{3}}}{d_s}$ and $F_r$, $\frac{d_{50}}{d_s}$, $\frac{v_c}{V}$, $\frac{v_a}{V}$. The independent variables were adopted for a non-uniform sediment size 0.24 mm at the side slit weir, with $R^2$ = 0.84, 0.985, 0.98 and 0.985, respectively.

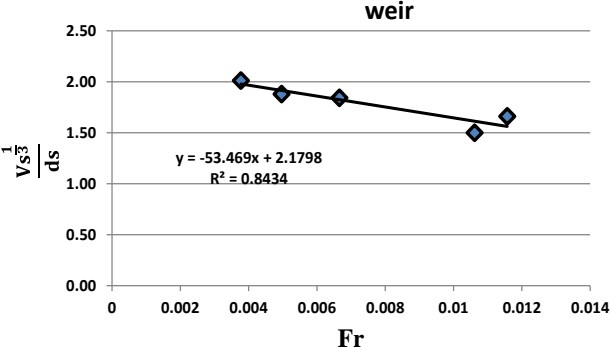

**Figure 1.** The relationship between the Froude number and $\frac{V_s^{\frac{1}{3}}}{d_s}$.

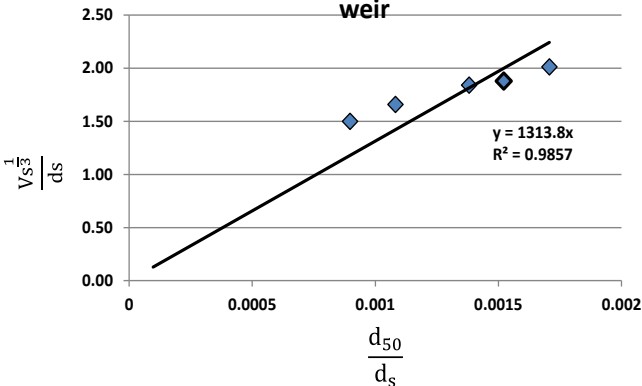

**Figure 2.** The relationship between $\frac{d_{50}}{d_s}$ and $\frac{V_s^{\frac{1}{3}}}{d_s}$.

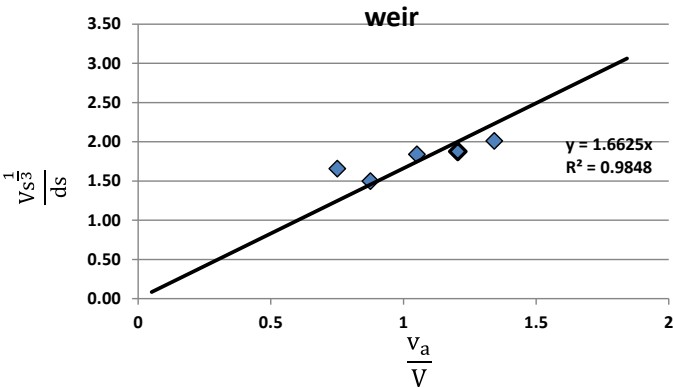

**Figure 3.** The relationship between $\frac{v_a}{V}$ and $\frac{V_s^{\frac{1}{3}}}{d_s}$.

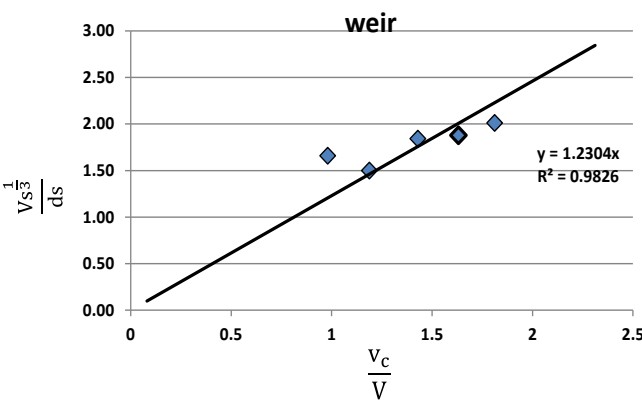

**Figure 4.** The relationship between $\frac{v_c}{V}$ and $\frac{V_s^{\frac{1}{3}}}{d_s}$.

## 3. The Experimental Work

The experiments were conducted using a rectangular flume comprising brick and concrete. The flume was designed and built in a hydraulic laboratory at the Ministry of Water Resources, Iraq. The flume was 8 m long, 1.25 m wide and 1 m deep. A slit weir was manufactured from steel (0.4 cm thick, 25 cm wide and 60 cm deep) and was fixed tightly 8 m from the flume inlet. Figure 5a–c show the flume and the arrangements used to conduct the experiments. The 2 m working section was filled with the uniform fine sediment ($d_{50}$ = 0.24 mm) up to 30 cm. Then, experiments were conducted on slit weirs located at the center and at the side of the flume, and the same experiments were repeated with uniform coarse sediment ($d_{50}$ = 0.55 mm). However, the same sets of experiments were conducted on non-uniform fine mobile beds and then on coarser mobile beds. It is essential to highlight that the sizes of both the uniform and non-uniform sediments were identical ($d_{50}$ = 0.24 mm and $d_{50}$ = 0.55 mm).

Before the commencement of the experiments, the flume and the working section were flooded with water in order to replace the air voids between the sediment with water. The flooding time was 1 h. For each sediment size, the discharges that were used were 125, 95.0, 62.0, 50.0 and 34.0 L/s, respectively, with a total of 44 test runs. The discharge in the flume was regulated by controlling the weir located at the flume upstream. The effect of water turbulence at the flume entrance on the mobile bed was eliminated by using the control basin.

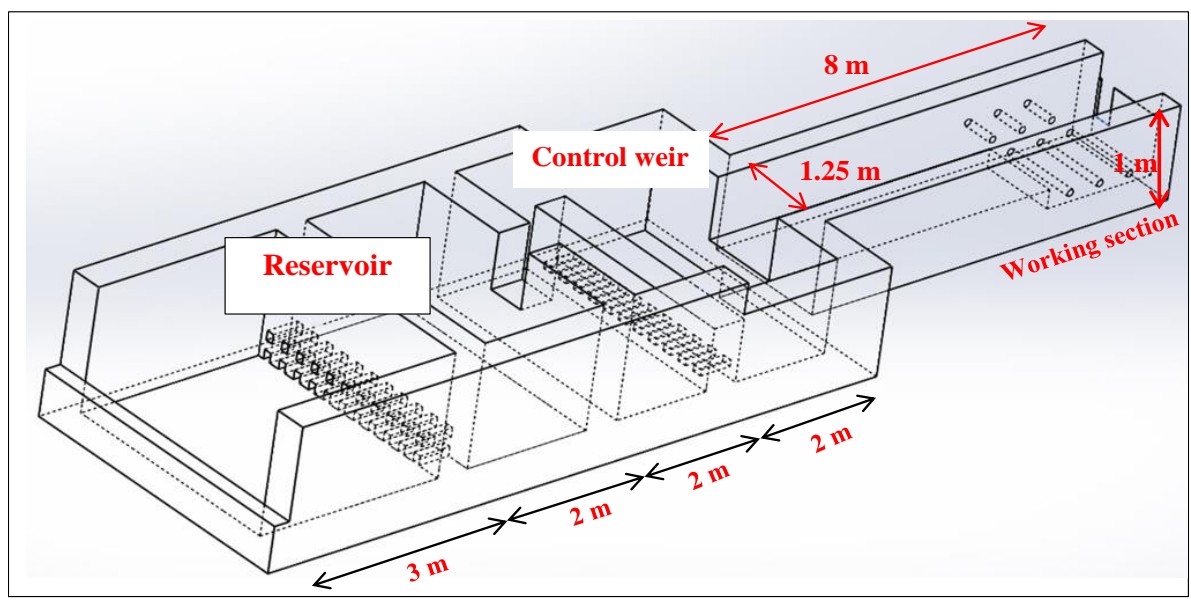

(**a**)

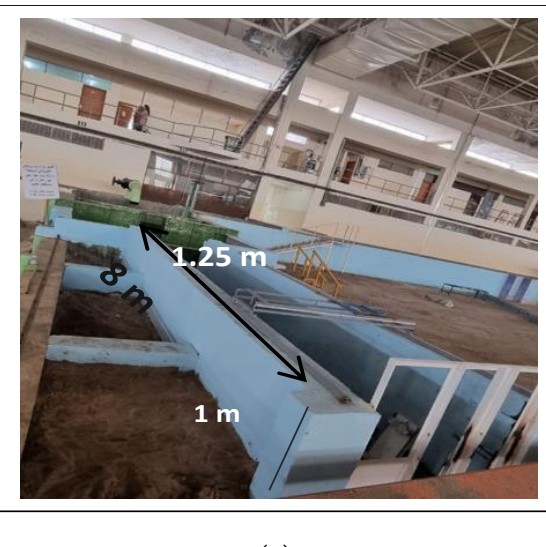

(**b**)          (**c**)

**Figure 5.** Laboratory flume: (**a**) Flume sketch; (**b**) Flume top view; (**c**) Flume side view.

The bed level measurements in the working section or mobile bed were carried out by using a point gauge, where the changes in bed level were monitored at the corners of 2 × 2 cm grids after 1, 2, 3, 4, 5, 6, 7 and 8 h from the commencement of each run. The measurements of velocity in 2D were conducted at various locations along the working section using the velocity measurement shown in Figure 6. The velocity was measured every 0.5 m along the flume sections. The average velocities of the flow condition were 0.26, 0.2, 0.17 and 0.15 m/s. The mechanical point gauge had an accuracy of ±1 mm (Figure 7), and it was used to measure the water level in the flume. In order to demonstrate the scour volume occurring upstream of the slit weir, the collected data on the bed level at the working section were used as input data for the Surfer program. The experimental design which describes more detail on the experiments is summarized in Table 2.

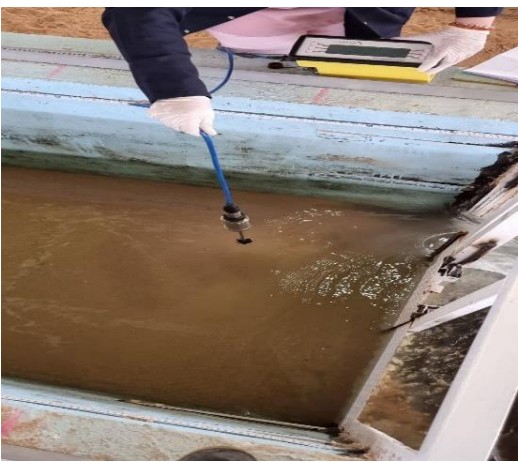

**Figure 6.** Two-dimensional velocity measurement.

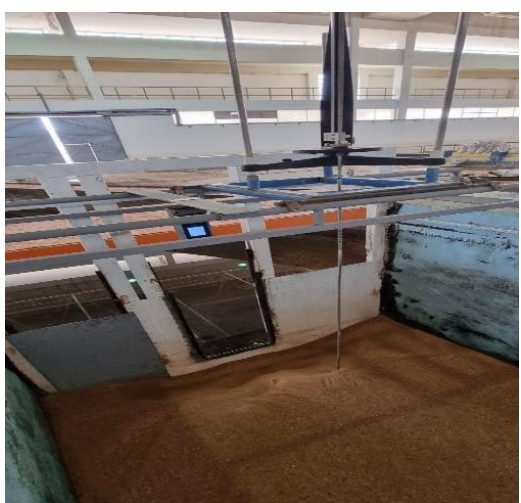

**Figure 7.** Mechanical point gauge.

**Table 2.** The experimental tests under steady and unsteady flow conditions.

| No. | Weir Location | Weir Dimensions Width (cm) × Height (cm) | Q (L/s) | Sediment Type | $d_{50}$ (mm) | No. of Runs |
|---|---|---|---|---|---|---|
| | | | **Steady condition** | | | |
| 1 | Center | 25 × 60 | 125.0, 95.0, 62.0, 50.0, and 34.0 | Uniform | 0.24 | 5 |
| 2 | Center | 25 × 60 | 125.0, 95.0, 62.0, 50.0, and 34.0 | Uniform | 0.55 | 5 |
| 3 | Center | 25 × 60 | 125.0, 95.0, 62.0, 50.0, and 34.0 | Non-uniform | 0.24 | 5 |
| 4 | Center | 25 × 60 | 125.0, 95.0, 62.0, 50.0, and 34.0 | Non-uniform | 0.55 | 5 |
| 5 | Side | 25 × 60 | 125.0, 95.0, 62.0, 50.0, and 34.0 | Uniform | 0.24 | 5 |
| 6 | Side | 25 × 60 | 125.0, 95.0, 62.0, 50.0, and 34.0 | Uniform | 0.55 | 5 |
| 7 | Side | 25 × 60 | 125.0, 95.0, 62.0, 50.0, and 34.0 | Non-uniform | 0.24 | 5 |
| 8 | Side | 25 × 60 | 125.0, 95.0, 62.0, 50.0, and 34.0 | Non-uniform | 0.55 | 5 |

**Table 2.** *Cont.*

| No. | Weir Location | Weir Dimensions Width (cm) × Height (cm) | Q (L/s) | Sediment Type | $d_{50}$ (mm) | No. of Runs |
|-----|---------------|------------------------------------------|---------|---------------|---------------|-------------|
| | | | **Unsteady condition** | | | |
| 9 | Center | 25 × 60 | 125.0, 62.0, 34.0 | Uniform and non-uniform | 0.24 | 2 |
| 10 | Center | 25 × 60 | 125.0, 62.0, 34.0 | Uniform and non-uniform | 0.55 | 2 |
| | | | **Total number of test runs** | | | 44 |

## 4. Results

To study the effect of sediment coarseness, the experiments were conducted using two types of uniform sand in the working section. One type had a median particle of $d_{50} = 0.24$ mm, and the other type had a median particle of $d_{50} = 0.55$ mm. However, to study the effect of uniformity on the scour volume upstream of the slit weir, uniform and non-uniform sand with the same above median particles ($d_{50} = 0.24$ mm and $d_{50} = 0.55$ mm) were used to run the experiments with different discharges, as mentioned previously.

The standard method (ASTM, 2006) was followed to carry out the sieve analysis for the sediment used in the experiments. The uniformity of the sand particles was determined from the grading curve after calculating the geometric standard deviation $\sigma_g$ using the following formula [31]:

$$\sigma_g = \sqrt{\frac{d_{84}}{d_{16}}} \tag{4}$$

For fine uniform and non-uniform sediment used in the mobile bed, the values of $d_{84}$, $d_{50}$ and $d_{16}$ were determined from the grading curves shown in Figures 8 and 9. However, Figures 10 and 11 show the grading curves for the coarser sediment, both uniform and non-uniform.

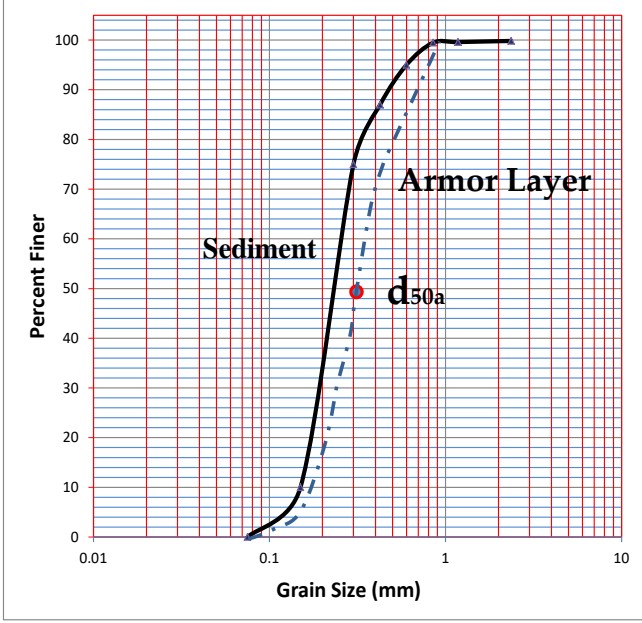

**Figure 8.** Grain size distribution curve for non-uniform sediment size $d_{50} = 0.24$ mm and armor layer.

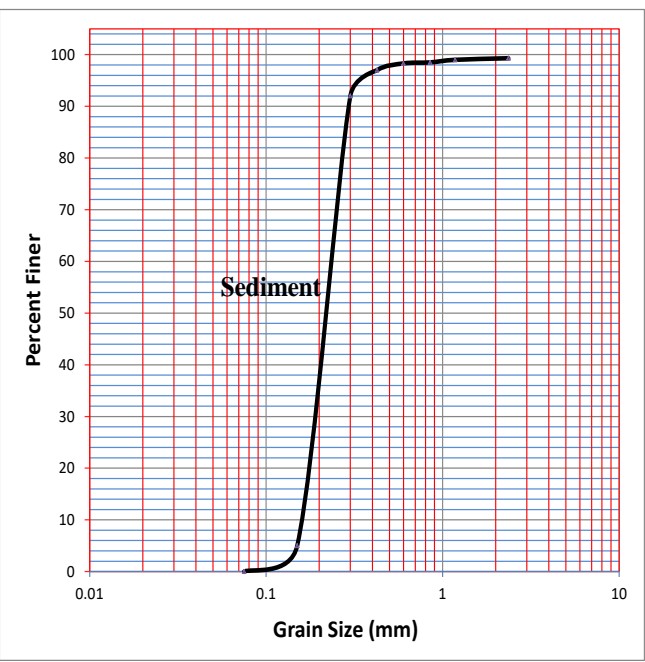

**Figure 9.** Grain size distribution curve for uniform sediment size $d_{50}$ = 0.24 mm.

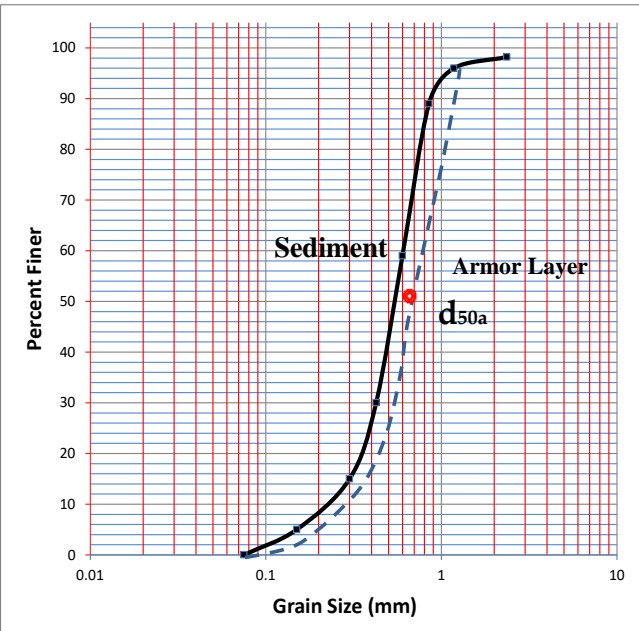

**Figure 10.** Grain size distribution curve for non-uniform sediment size $d_{50}$ = 0.55 mm and armor layer.

Table 3 shows the values of the geometric standard deviation ($\sigma_g$) for the two types of the selected sand. For uniform sediment with $d_{50}$ = 0.24 mm, the value of $\sigma_g$ was 1.28, and for the sediment with $d_{50}$ = 0.55 mm, $\sigma_g$ was 1.26. However, for non-uniform sediment sizes 0.24 and 0.55 mm, the values of $\sigma_g$ were 1.55 and 1.6, respectively. The values of $d_{max}$ and $d_{min}$ were extracted from sediment grading curves. The following equation was used to determine $d_{50a}$ [31]:

$$d_{50a} = \frac{d_{max}}{1.8} \tag{5}$$

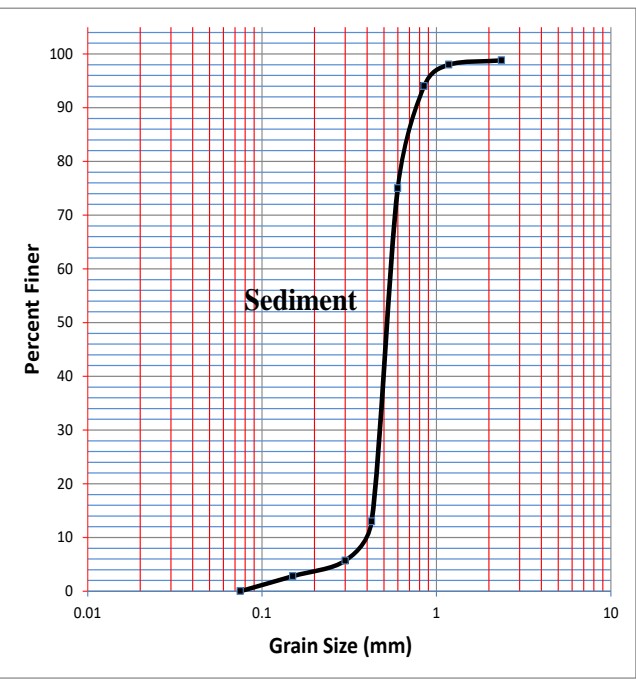

**Figure 11.** Grain size distribution curve for uniform sediment size $d_{50}$ = 0.55 mm.

**Table 3.** Sedimentation properties.

| Sand Type | $d_{84}$ (mm) | $d_{50}$ (mm) | $d_{16}$ (mm) | $\sigma_g$ | $d_{max}$ (mm) | $d_{50a}$ (mm) | Bulk Density (kg/m³) |
|---|---|---|---|---|---|---|---|
| Uniform sand | 0.28 | 0.24 | 0.17 | 1.28 | 0.3 | - | 1349 |
| Uniform sand | 0.72 | 0.55 | 0.45 | 1.26 | 0.8 | - | 1436 |
| Non-uniform sand | 0.4 | 0.24 | 0.16 | 1.55 | 0.5 | 0.3 | 1315 |
| Non-uniform sand | 0.81 | 0.55 | 0.31 | 1.6 | 0.9 | 0.5 | 1518 |

In the present study, $d_{90}$ was adopted as $d_{max}$ and $d_{50a}$, and the values are presented in Table 3.

*4.1. Clear Water Scour*

Clear water scour occurs when there is no transport of bed particles from the scour hole upstream of the slit weir. In clear water scour conditions, the shear stress is either equal to or less than the critical shear stress for the sediment initiation of motion, and the flow velocity (v) is less than the threshold velocity ($v_c$). Thus, ref. [31] specifies that the clear water scour conditions for the flow intensity should be $v/v_c < 1$ when $\sigma_g < 1.3$ for uniform sediment. In this study, the flow intensities were 0.71 and 0.84 for sediment sizes of $d_{50}$ = 0.55 and 0.24 mm, respectively. $(v - (v_a - v_c))/v_c < 1$ and $\sigma_g > 1.3$ for non-uniform sediment, which was 0.92 for a sand size of $d_{50}$ = 0.55 mm and 0.9 for a sand size of $d_{50}$ = 0.24 mm. In these flow conditions, the armor layer reduces the sour depth value, and the flow intensity ratio is replaced by $v/v_a$.

The critical approach flow velocity for sediment bed entrainment and armor peak velocity were determined based on shield diagram equations [8]:

$$v_c = 0.049 + 0.053\ d_{50}^{1.4} + \left(0.066 + 0.072\ d_{50}^{1.4}\right) \log \frac{y}{d_{50}} \text{ for 0.1mm } < d_{50} < 1\text{mm} \qquad (6)$$

$$v_a = 0.039 + 0.018\ d_{90}^{1.4} + \left(0.052 + 0.025\ d_{90}^{1.4}\right) \log \frac{y}{d_{90}} \text{ for 0.1 mm } < 0.55 d_{90} < 1 \text{ mm} \quad (7)$$

Figure 12 shows the procedure that was used to classify the sediment type in the mobile bed.

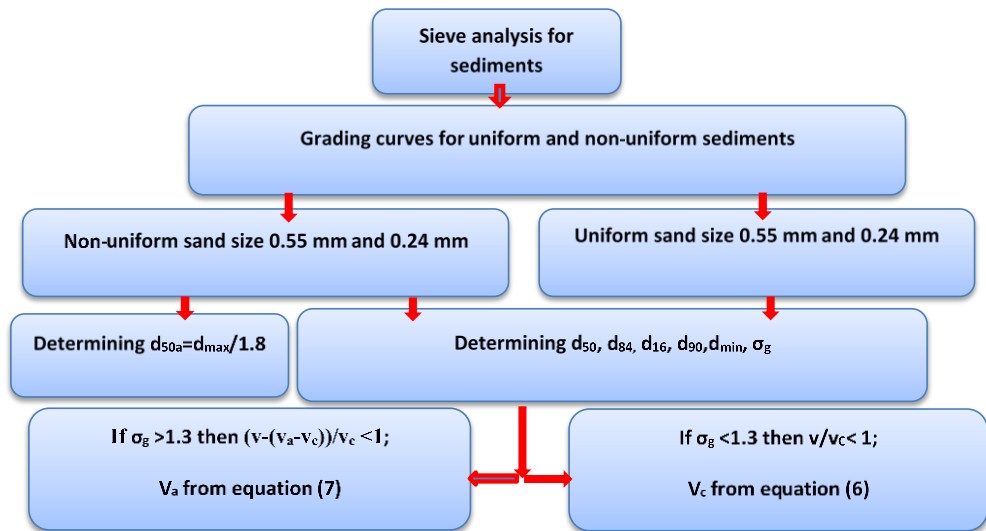

**Figure 12.** A Procedure used for the determination of $v_c$ for uniform sediment and $v_c$ and $v_a$ for non-uniform sediment.

Furthermore, the maximum scour volume can be reached when there are no addition sediment particles removed from the scour hole by the flowing water. All experimental tests in this study were carried out until an equilibrium state was reached after 28,800 s. The scour hole development upstream of the slit weir located at the center of the flume is shown in Figure 13.

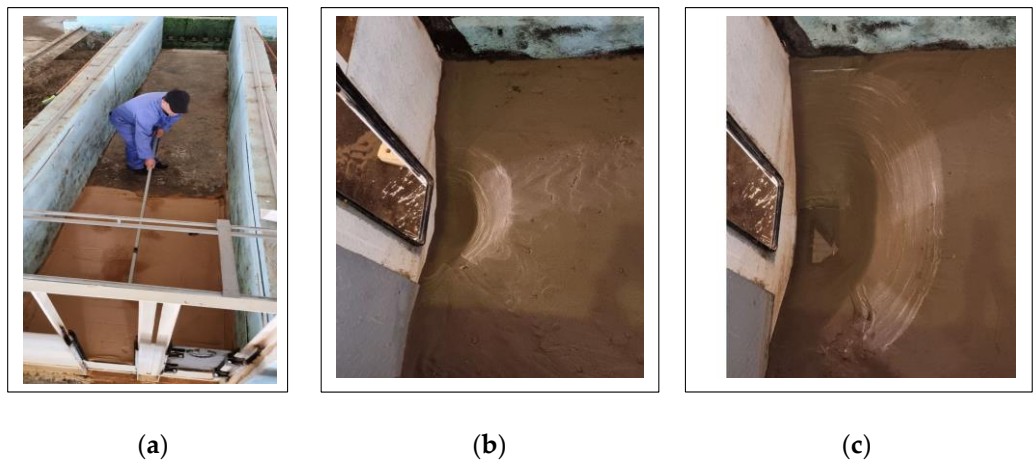

| (a) | (b) | (c) |

**Figure 13.** Scour hole development upstream of the center slit weir: (**a**) Leveling working section; (**b**) Initial scour hole; (**c**) Final scour hole reaching equilibrium conditions.

### 4.2. Equilibrium Scour Time

Based on [6], this research, which investigated the scour upstream of a slit weir with uniform sedimentation, recommends examination of the effect of sand non-uniformity on a scour volume. Therefore, a comparison was conducted between the maximum scour volume resulting at the upstream center and side slit weirs for uniform and non-uniform sediment for sand sizes of 0.24 and 0.55 mm. The tests were carried out with a maximum flow rate of 125 L/s, as presented in Figures 14–17, in order to investigate the effect of the sediment size and uniformity for the selected sand, which had a $\sigma_g$ of 1.28 for uniform sand and 1.55 for a non-uniform sand size of $d_{50} = 0.24$ mm and a $\sigma_g$ of 1.26 for uniform sand and 1.6 for a non-uniform sand size of $d_{50} = 0.55$ mm. The scour hole reached equilibrium conditions after 8 h from the commencement of the experiment. The maximum scour volumes were measured with a uniform sand size of $d_{50} = 0.24$ mm at the center slit weir,

which was 1.1 times the one recorded for the non-uniform sediment for the same sand size. In addition, this value differs by 1.4 for the scour volume when adopting a sand size of $d_{50} = 0.55$ mm. When the slit was located on the side of the weir and the sand size was $d_{50} = 0.24$ mm, the value of the scour volume was 1.3 times than that of the non-uniform sand. The resulting scour volume for a sand size of $d_{50} = 0.55$ mm on the upstream side of the slit weir was 24% higher than the value recorded with non-uniform sand. It is obvious in this study that the scour volume had smaller values when the sand non-uniformity increased, as mentioned in [6].

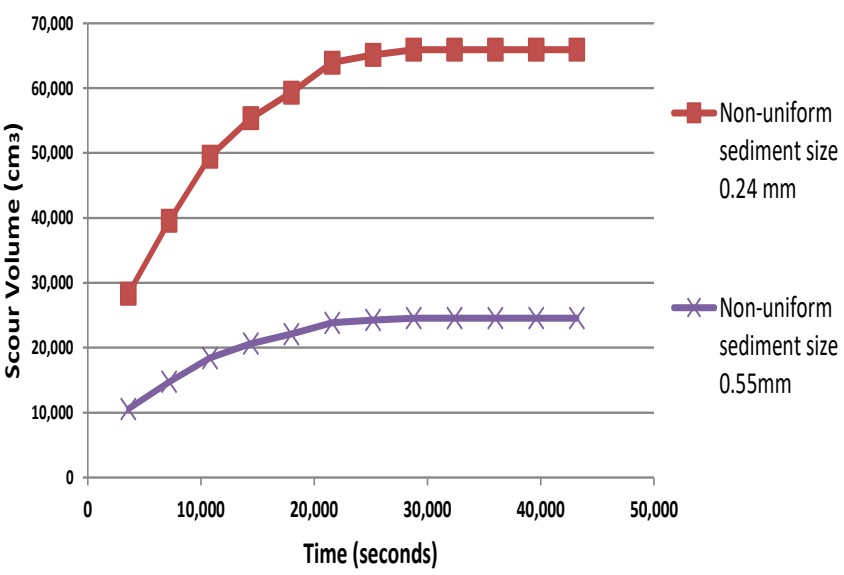

**Figure 14.** Time varying scour volume for non-uniform sediment sizes of 0.24 mm and 0.55 mm at the center slit weir.

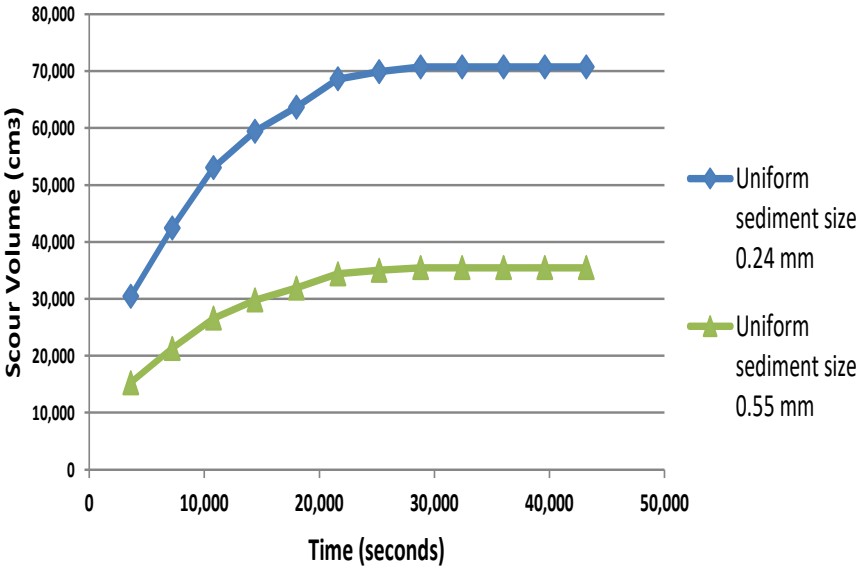

**Figure 15.** Time varying scour volume for uniform sediment sizes of 0.24 mm and 0.55 mm at the center slit weir.

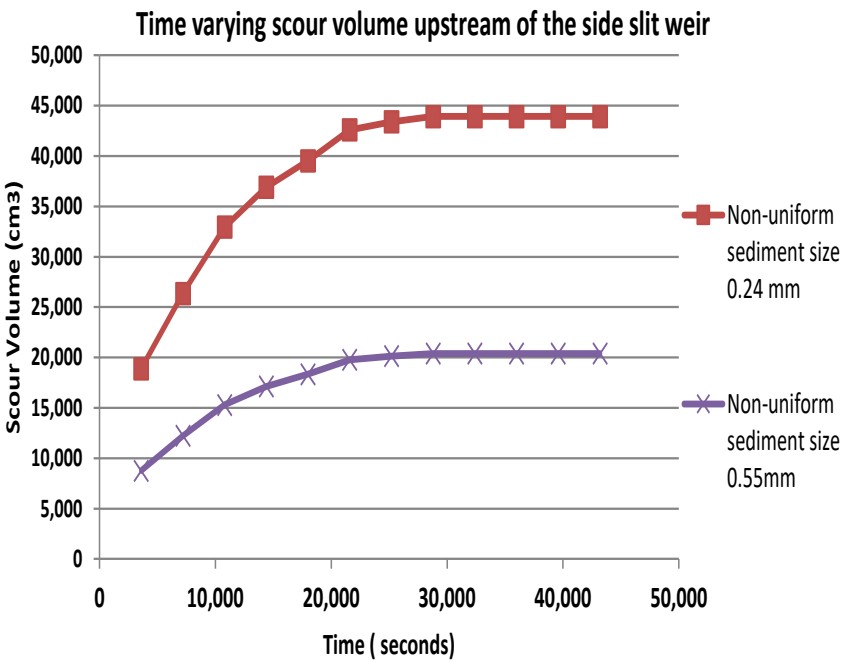

**Figure 16.** Time varying scour volume for non-uniform sediment sizes of 0.24 mm and 0.55 mm at the side slit weir.

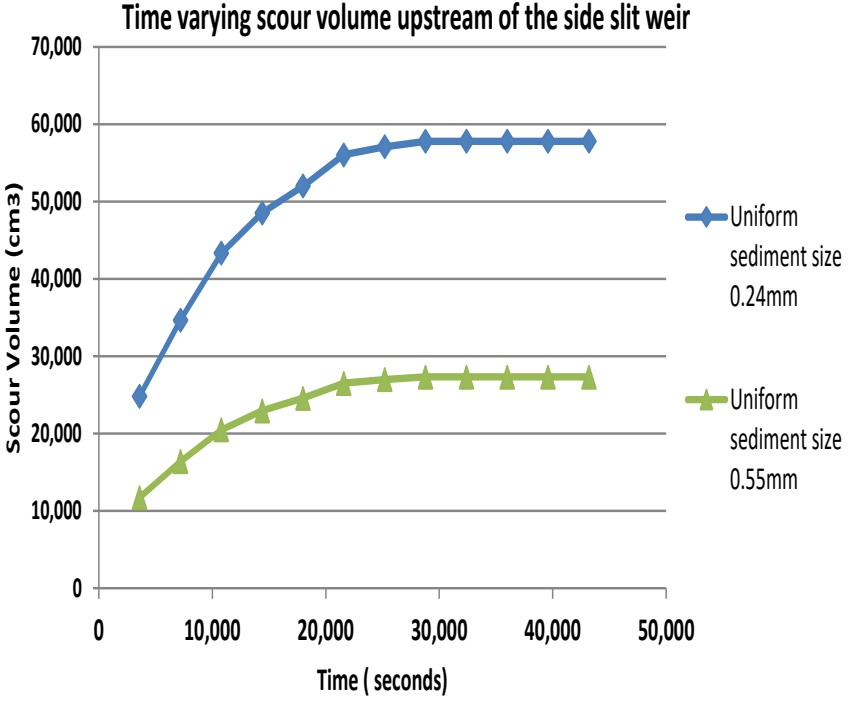

**Figure 17.** Time varying scour volume for uniform sediment sizes of 0.24 mm and 0.55 mm at the side slit weir.

### 4.3. Mechanism of Scour Hole Development and Velocity Distribution

The formation of vortices upstream of the slit weir in the scour hole made the shear stress exceeds the critical value. The downflow rolled up and continued to create a hole due to the interaction with the oncoming flow, developing into a complex vortex and moving from the side to the center of the scour hole with different intensities and radii. Later, the generated vortices moved all over the scour hole with time-varying intensities. This condition continued until the scour hole reached equilibrium conditions, re-

sulting in a maximum scour volume. The vortices entrained sediment from the bed within the vicinity of the silt weir and carried it out through the slit opening. The dominant affecting parameter on scour progress was the flow rate as well as the sediment size and sediment uniformity; thus, when the flow rate increased, it then increased the shear stress, leading to enlarging the scour hole dimensions for the same sediment size and type.

Figures 18 and 19 show the 2D and 3D presentations of velocity up to a distance of 6 m from the weir with the slit at the center and the slit on the side of the flume. The velocity was measured every 0.50 m starting from the upstream side of the flume toward the slit weir opening, and the last section of the measurement was located 0.40 m upstream of the weir slit. When the slit was located at center of the weir, the flow close to the slit opening tended upward due to the high value of shear stress, because the maximum velocity was recorded at the center of the open flow pathway, which generated high kinetic energy turbulence and created rotating vortices upstream of the slit weir, causing a local scour hole and releasing the sediment accumulation around the slit, as shown in Figure 20a–c. In contrast to both sides of the flume, the flow tended downward because the measured minimum values of velocity caused weaker vortices. This clarifies the reason that the maximum scour volume was observed when the slit was located at center of the weir.

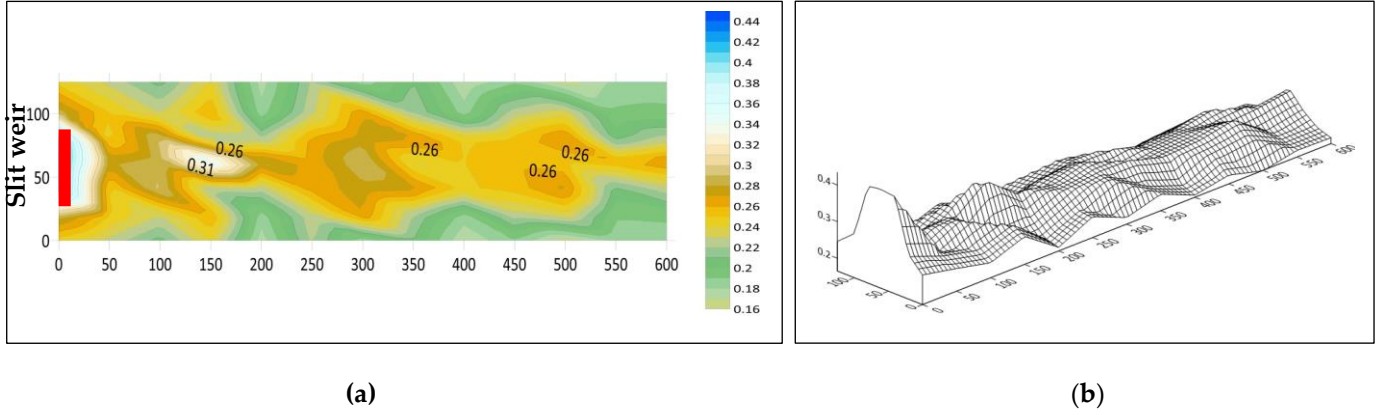

**(a)**                                                                                     **(b)**

**Figure 18.** Velocity distribution along the flume: (**a**) Velocity contour lines up to 6 m upstream of the center slit weir; (**b**) Three-dimensional velocity distribution.

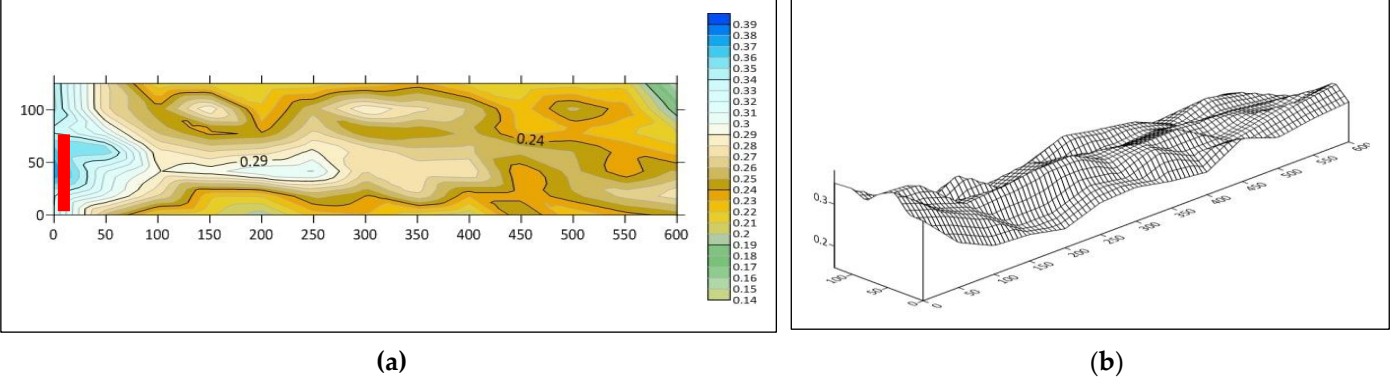

**(a)**                                                                                     **(b)**

**Figure 19.** Velocity distribution along the flume: (**a**) Velocity contour lines up to 6 m upstream of the side slit weir; (**b**) Three-dimensional velocity distribution.

The maximum scour area dimensions are presented in Tables 4 and 5 according to the flow intensities and slit weir locations as well as the sediment uniformity for both sand sizes of $d_{50}$ = 0.24 mm and $d_{50}$ = 0.55 mm. Moreover, Figures 21–24 show the comparison between the maximum scour volume under different flow intensities of 125, 95, 62, 50 and 34 L/s for sediment sizes of 0.24 and 0.55 mm with uniform and non-uniform sediment upstream of the center and side slit weirs.

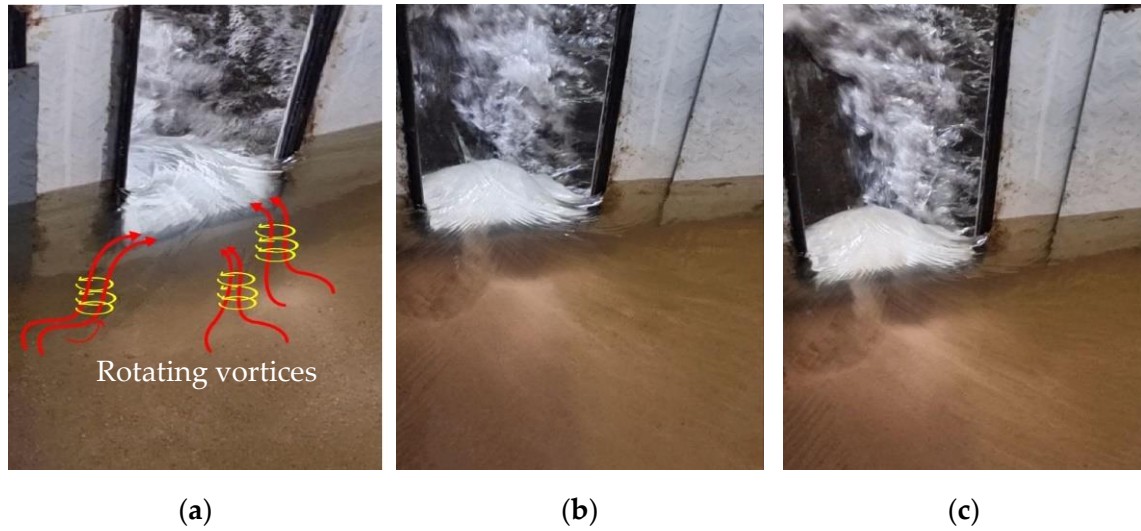

**(a)**         **(b)**         **(c)**

**Figure 20.** Scour by vortex action: (**a**) Initial stage; (**b**,**c**) Rotating vortices with sediment release.

**Table 4.** Scour area at the side and center slit weirs for a sand size of $d_{50}$ = 0.24 mm.

| No. | Scenario | Side Opening $d_{50}$ = 0.24 mm | | Center Opening $d_{50}$ = 0.24 mm | |
|---|---|---|---|---|---|
| | Flat crest | Uniform sand Scour hole dimensions (x × y) cm | Non-uniform sand Scour hole dimensions (x × y) cm | Uniform sand Scour hole dimensions (x × y) cm | Non-uniform sand Scour hole dimensions (x × y) cm |
| 1 | Q = 125 L/s | 60 × 70 | 50 × 70 | 60 × 110 | 57 × 110 |
| 2 | Q = 95 L/s | 40 × 50 | 40 × 50 | 56 × 110 | 47 × 110 |
| 3 | Q = 62 L/s | 40 × 50 | 40 × 50 | 42 × 90 | 40 × 80 |
| 4 | Q = 50 L/s | 40 × 40 | 40 × 40 | 53 × 90 | 40 × 70 |
| 5 | Q = 34 L/s | 40 × 40 | 32 × 40 | 40 × 70 | 35 × 60 |

**Table 5.** Scour area at the side and center slit weirs for a sand size of $d_{50}$= 0.55 mm.

| No. | Scenario | Side Opening $d_{50}$ = 0.55 mm | | Center Opening $d_{50}$ = 0.55 mm | |
|---|---|---|---|---|---|
| | Flat crest | Uniform sand Scour hole dimensions (x × y) cm | Non-uniform sand Scour hole dimensions (x × y) cm | Uniform sand Scour hole dimensions (x × y) cm | Non-uniform sand Scour hole dimensions (x × y) cm |
| 1 | Q = 125 L/s | 37 × 70 | 26 × 70 | 40 × 100 | 40 × 100 |
| 2 | Q = 95 L/s | 37 × 70 | 25 × 60 | 40 × 100 | 35 × 100 |
| 3 | Q = 62 L/s | 36 × 50 | 25 × 60 | 40 × 90 | 35 × 90 |
| 4 | Q = 50 L/s | 36 × 50 | 24 × 50 | 40 × 90 | 32 × 90 |
| 5 | Q = 34 L/s | 32 × 50 | 21 × 50 | 32 × 70 | 24 × 70 |

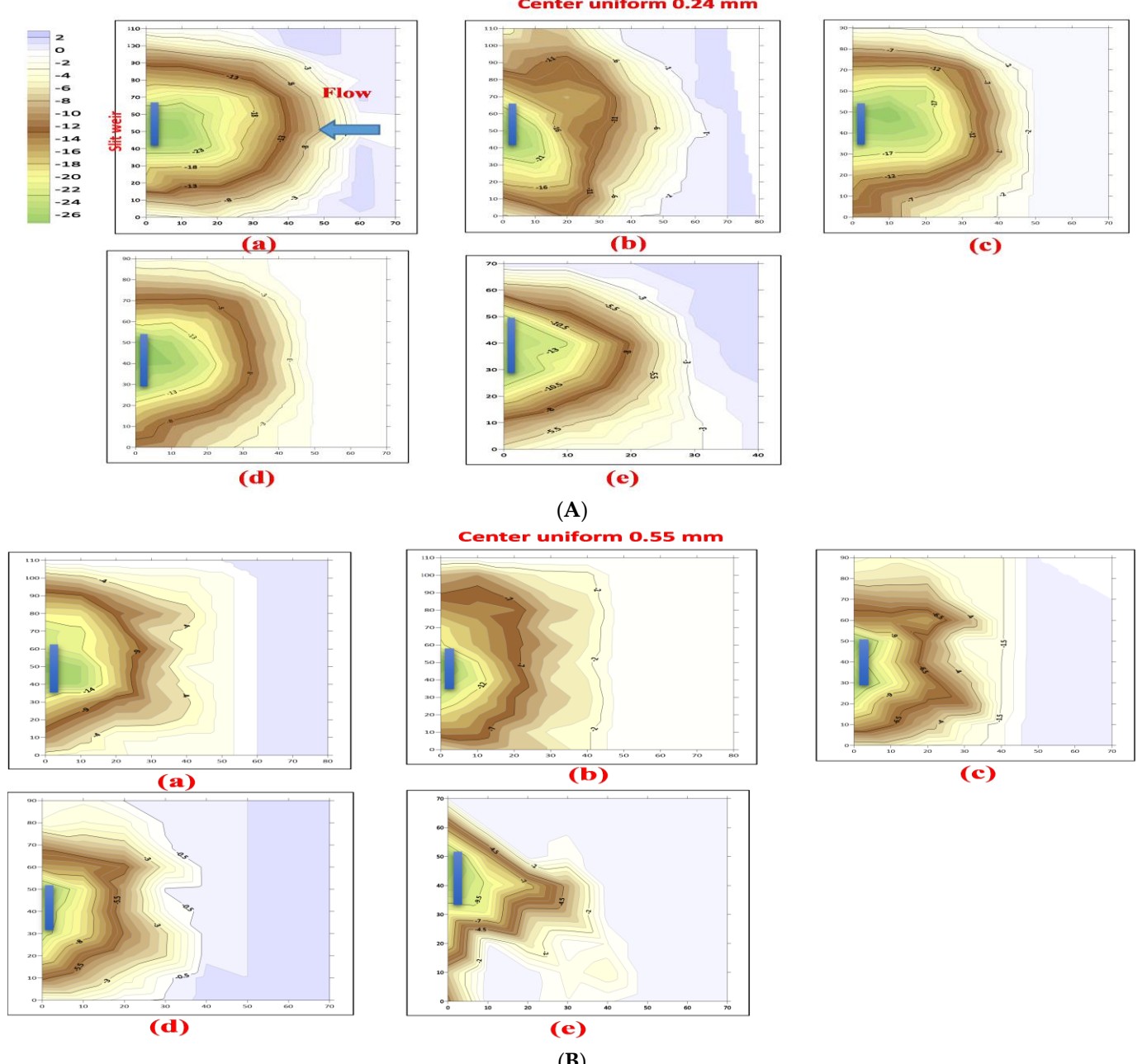

**Figure 21.** Scour hole contour lines under different flow intensities with the center slit weir: (**A**) 0.24 mm; (**B**) 0.55 mm: (a) Flow rate = 125 L/s; (b) Flow rate = 95 L/s; (c) Flow rate = 62 L/s; (d) Flow rate = 50 L/s; (e) Flow rate = 34 L/s.

This demonstrates that the resulting scour volume with a uniform sediment size of $d_{50}$ = 0.24 mm with the slit located at center of the weir was reduced by 70%, and it was 78% for a sand size of $d_{50}$ = 0.55 mm when the flow rate changed from 125 L/s to 34 L/s. There was a 75% reduction for a non-uniform sand size of $d_{50}$ = 0.24 mm and a 73% reduction for a sand size of $d_{50}$ = 0.55 mm. The resulting scour volume at the side slit weir was minimized by 69% for a uniform sand size of $d_{50}$ = 0.24 mm and by 77% for a sand size of $d_{50}$ = 0.55 mm. However, the scour volume was reduced by 65% for a non-uniform sediment size of $d_{50}$ = 0.24 mm and by 79% for a sediment size of $d_{50}$ = 0.55 mm.

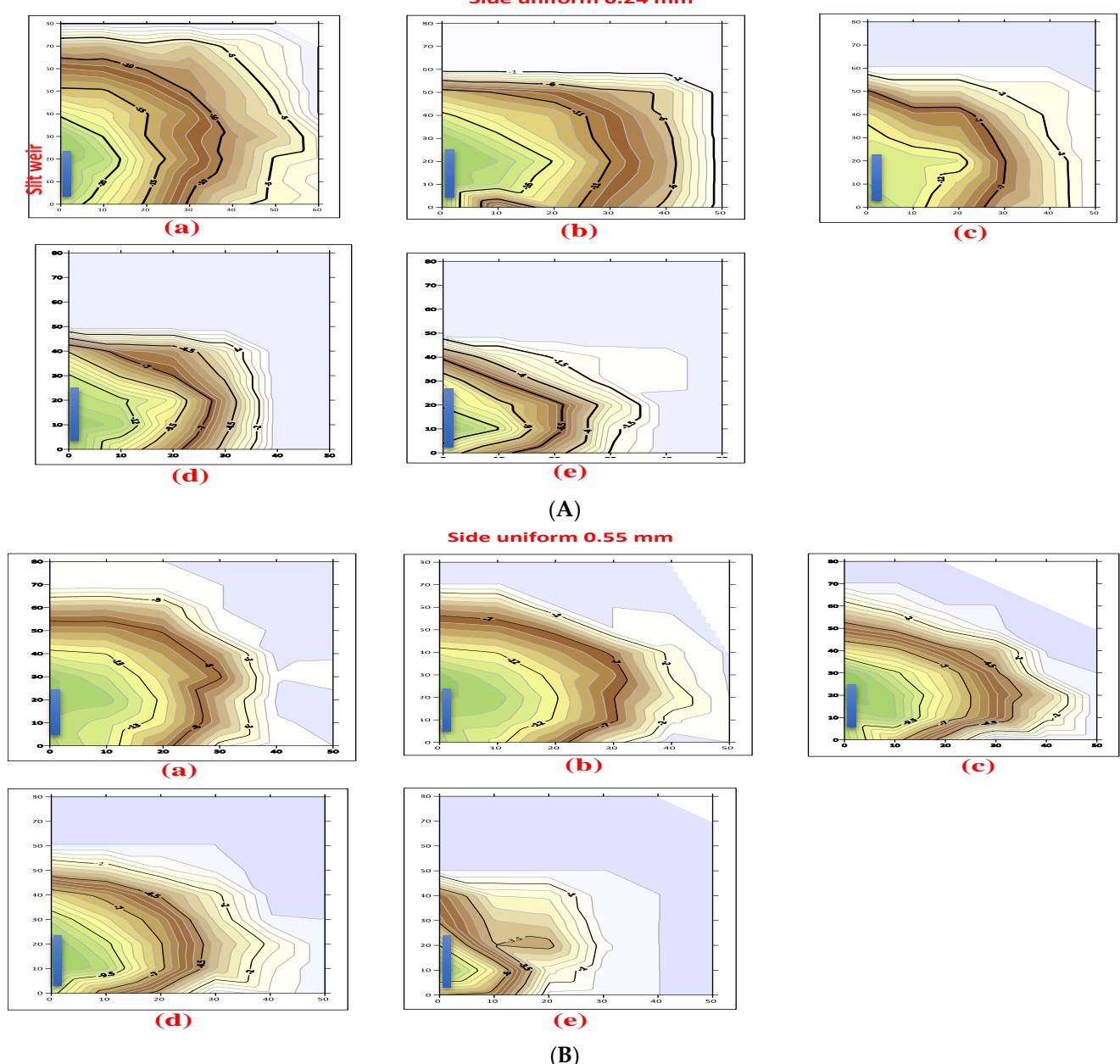

**Figure 22.** Scour hole contour lines under different flow intensities with the side slit weir: (**A**) 0.24 mm; (**B**) 0.55 mm: (a) Flow rate = 125 L/s; (b) Flow rate = 95 L/s; (c) Flow rate = 62 L/s; (d) Flow rate = 50 L/s; (e) Flow rate = 34 L/s.

The behavior of uniform and non-uniform sediment under unsteady flow conditions was tested for both sand sizes of 0.24 and 0.55 mm with the slit located at the center of weir. Figures 25 and 26 demonstrate the maximum scour volume investigated with an increasing flow rate in a gradual step from 34 and 62 to 125 L/s and decreasing in a same manner. It was observed that the maximum scour volume resulted in a peak flow rate and continued developing even though the flow rate values decreased until equilibrium conditions were achieved. The resulting value of the maximum scour volume for a sand size of $d_{50}$ = 0.24 mm was three times that of the uniform sediment for a sand size of $d_{50}$ = 0.55 mm, and triple values of the scour volume were recorded for non-uniform sediment with the same median size. The behavior of the maximum scour volume was proportioned positively with the uniformity of the sand and negatively with its size.

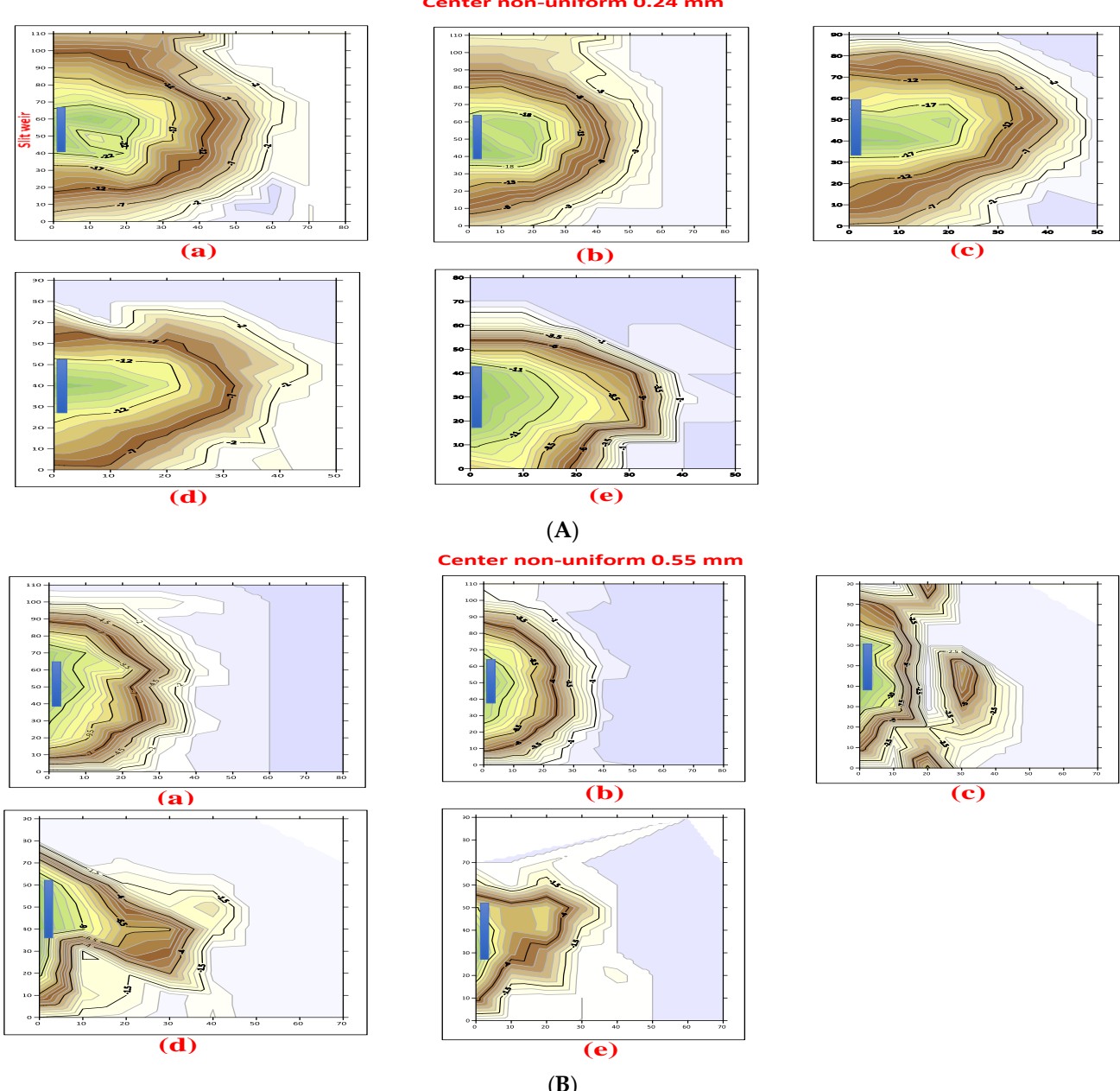

**Figure 23.** Scour hole contour lines under different flow intensities for non-uniform sediment of $d_{50}$ = 0.24 and 0.55 mm with the center slit weir: (**A**) 0.24 mm; (**B**) 0.55 mm: (a) Flow rate = 125 L/s; (b) Flow rate = 95 L/s; (c) Flow rate = 62 L/s; (d) Flow rate = 50 L/s; (e) Flow rate = 34 L/s.

The maximum resulting scour depths of this study and previous studies of scour around different hydraulic structures are presented in Table 6, which clarifies the experiment's boundary conditions and flow properties as well as the structure geometry configurations.

According to [6], Figures 27 and 28 show the relation between the maximum scour depth and scour volume based on [6]. Equation (8) is presented below for non-cohesive uniform sand. The tested data for the recent study were obtained from experimental measurements of the scour volume and scour depth for different flow rates of 125.0, 95.0, 62.0, 50.0, 34.0 L/s, and $0.2 \leq b_{sl}/B \leq 0.3$. Uniform and non-uniform sands with median particle sizes of $d_{50}$ = 0.24 mm and $d_{50}$ = 0.55 mm were adopted, as well as $q_{sl}/q < 5$, $v/v_c$ and $(v - (v_a - v_c)/v_c) < 1$. The scour volume was measured for different slit locations (center and side). B is the channel width, $b_{sl}$ is the slit weir width, q is the flow rate per unit

of width and q$_{sl}$ is the flow passing through the slit weir per unit of the weir slit width. The slope between V$_s$. and d$_s$ was found to be three for uniform and non-uniform sediment.

$$\frac{d_s}{V_s^{\frac{1}{3}}} = 0.39\left(\frac{b_{sl}}{B}\right)^{-0.383} \tag{8}$$

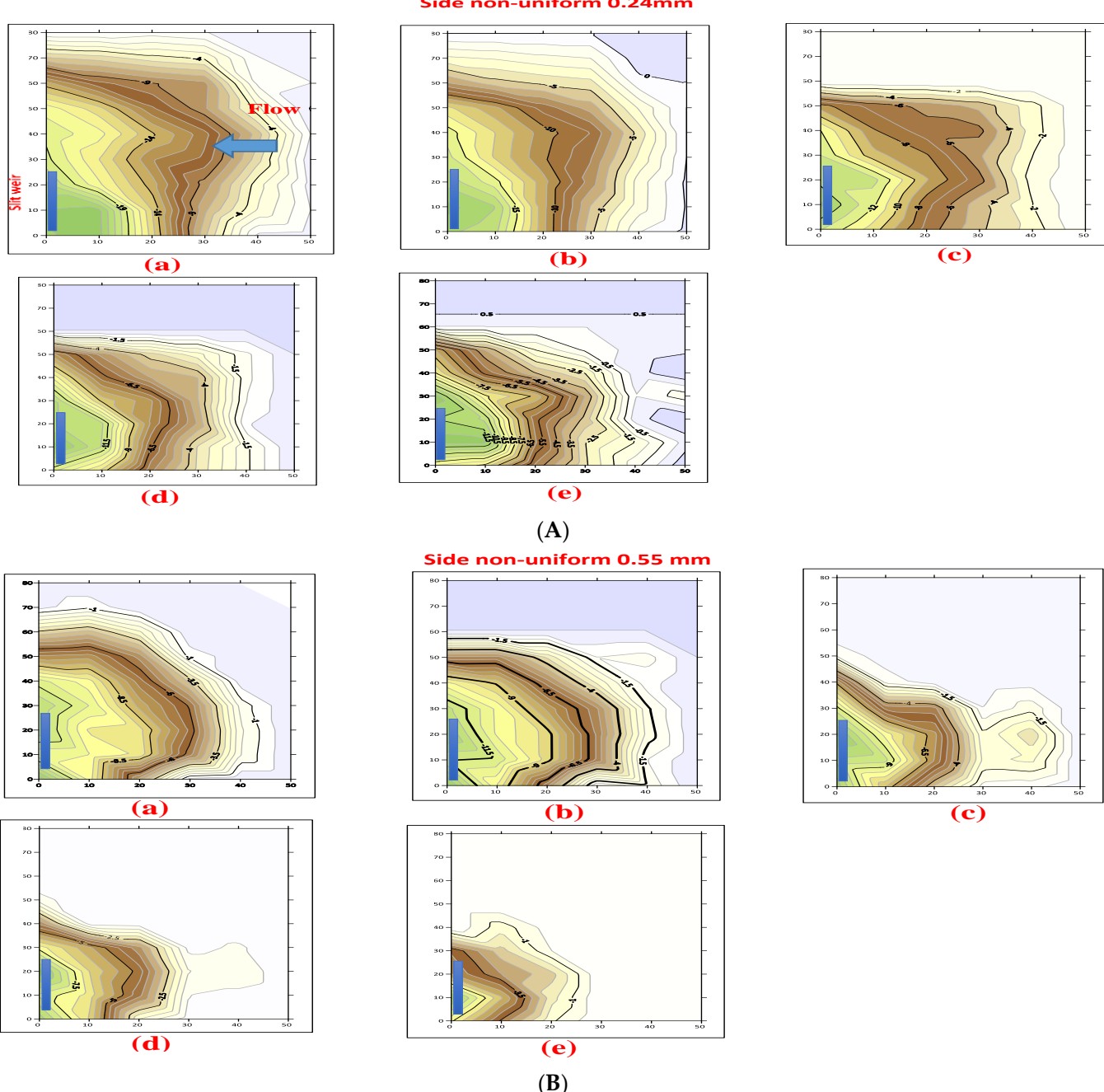

**Figure 24.** Scour hole contour lines under different flow intensities for non-uniform sand of d$_{50}$ = 0.24 and 0.55 mm with the side slit weir: (**A**) 0.24 mm; (**B**) 0.55 mm: (a) Flow rate = 125 L/s; (b) Flow rate = 95 L/s; (c) Flow rate = 62 L/s; (d) Flow rate = 50 L/s; (e) Flow rate = 34 L/s.

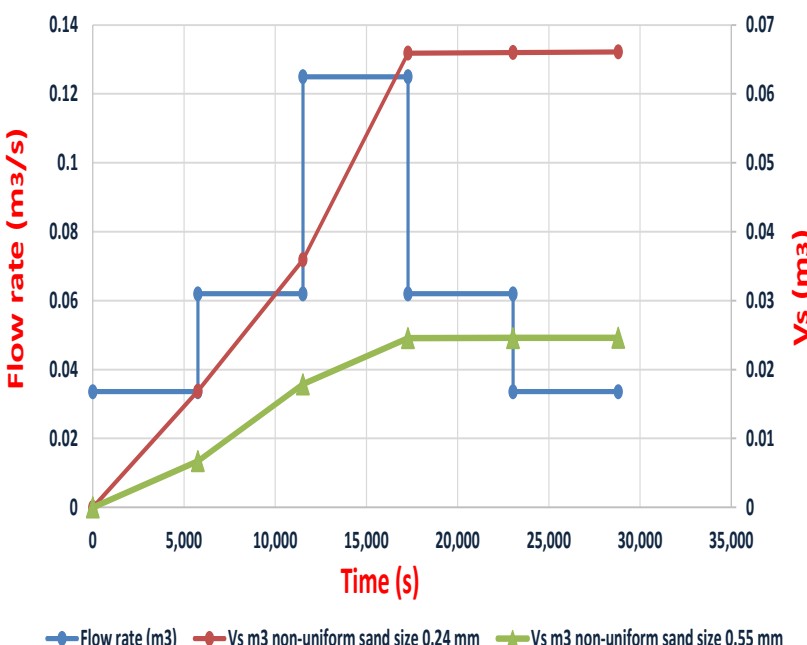

**Figure 25.** Unsteady flow conditions for non-uniform sediment sizes of $d_{50}$ = 0.24 mm and 0.55 mm at the center slit weir.

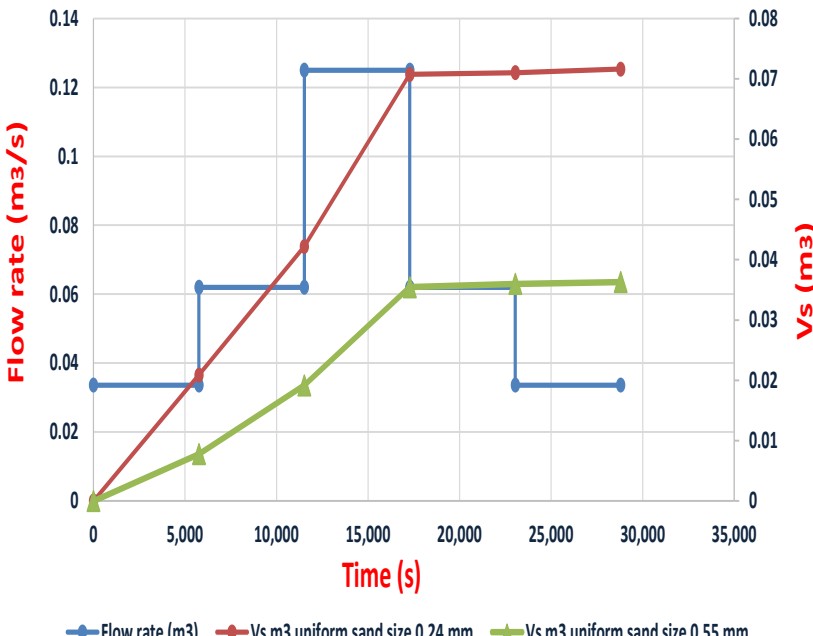

**Figure 26.** Unsteady flow conditions for uniform sediment sizes of $d_{50}$ = 0.24 mm and 0.55 mm at the center slit weir.

Even though [6] recommend to use Equation (8) with uniform sediment. Nevertheless, the predicted values of maximum scour volume for present study for uniform and non-uniform sediment had been estimated based on Equation (8) and compared within the values obtained and measured from the experimental work as shown in Figures 29 and 30. The accuracy between the predicted scour volume and the measured values was estimated by $R^2$ thus, $R^2$ was 0.89 for uniform sediment and 0.85 for non-uniform sediment.

**Table 6.** Summary of different studies on scour around hydraulic structures.

| | Gaudio [13] | Marion [14] | Guan [10] | Wang [11] | Ota [6] | Nkad [8] | This Study |
|---|---|---|---|---|---|---|---|
| Type of structure | Bed sills | Bed sills | Submerged weir | Submerged weir | Slit weir | Slit weir | Slit weir |
| Flowrate L/s | 45–81 | 18, 22, 26 | 15–86 | 12–89.3 | 7.2 | 2.6–8 | 34–125 |
| Sediment size $d_{50}$ mm | 4.1, 8.5/uniform | 8.7 uniform | 0.26, 0.85 uniform | 0.26, 0.85 uniform | 0.22, 0.77, 1/uniform | 0.3, 0.7 non-uniform | 0.24, 0.55 uniform and non-uniform |
| Number of tests | 19 | 48 | 37 | 62 | 33 | 6 | 44 |
| Flume dimensions m | 2.44 wide and 0.6 deep | 10 long, 0.5 wide and 0.5 deep | 12 long, 0.44 wide and 0.58 deep | 12 long, 0.44 wide and 0.38 deep | 10.5 long, 0.5 wide and 0.35 deep | 12 long, 0.3 wide and 0.3 deep | 8 long, 1.25 wide and 1 deep |
| Bed slope% | 0.0120–0.0018 | 0.042–0.041 | 0.0009–0.008 | 0.0004–0.0074 | Flat bed | Flat bed with ramp of 1:10 | Flat bed |
| Flow depth cm | 9–14 | - | 12–17.4 | 15–18 | 3.6–18 | 11.1–21.1 | 18–38 |
| Flow velocity m/s | 0.69–0.921 | Specific energy = water depth + $\frac{v^2}{2g}$ 7.8–10.1 | 0.281–1.124 | 0.185–1.166 | 0.13 | 0.05–0.240 | 0.15–0.26 |
| Fr | 0.64–0.95 | 1.06–1.44 | 0.13–0.88 | 0.13–0.89 | 0.17 | 0.04–0.16 | 0.112–0.148 |
| Maximum scour depth cm | 25.8 | 28.8 | 15.5 | 16.5 | 9.2 | 4.2 | 27 |

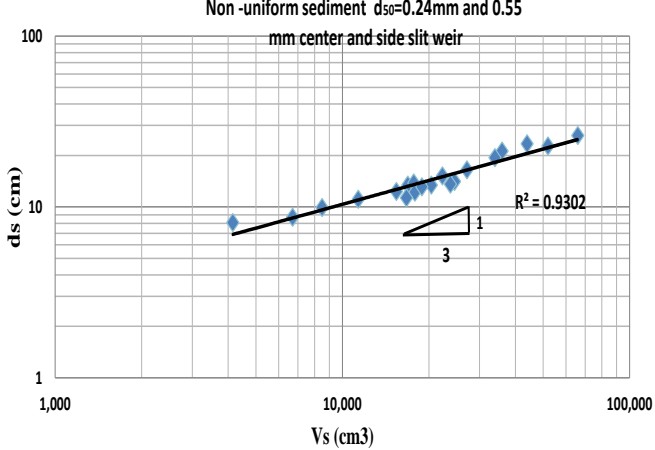

**Figure 27.** Relationship between maximum scour depth and volume for non-uniform sediment sizes of $d_{50}$ = 0.24, 0.55 mm.

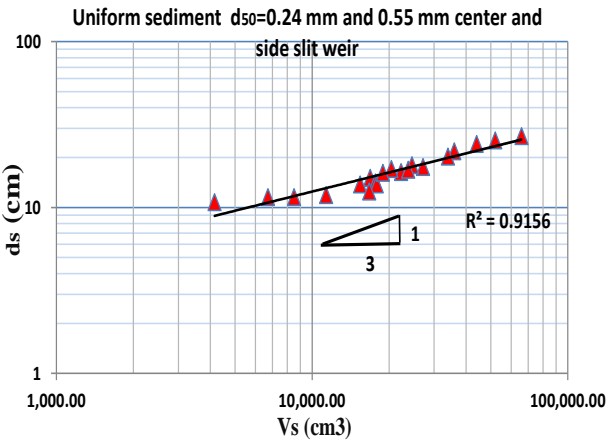

**Figure 28.** Relationship between maximum scour depth and volume for uniform sediment sizes of $d_{50}$ = 0.24, 0.55 mm.

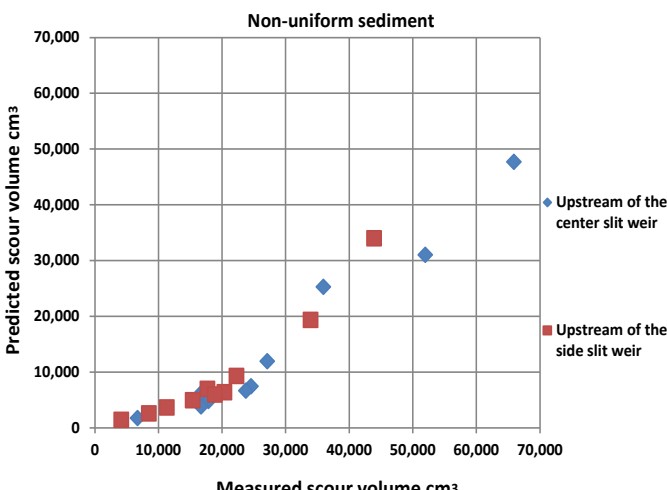

**Figure 29.** Validation of Equation (8) for non-uniform sediment.

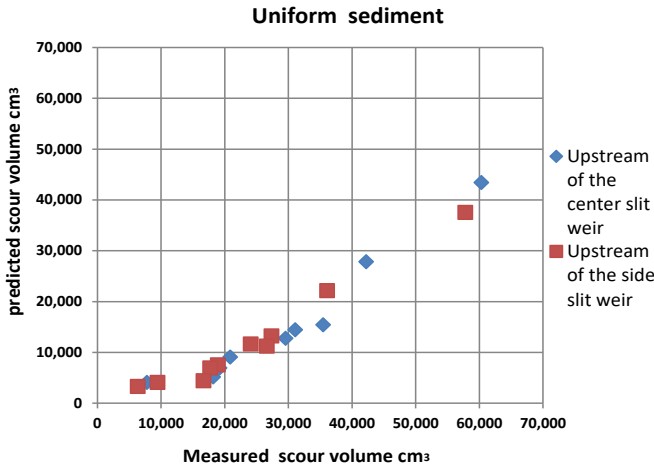

**Figure 30.** Validation of Equation (8) for uniform sediment.

## 5. Conclusions

The physical model and experimental work proposed in the present study were used to investigate the effect of flow intensity, sediment coarseness, sediment uniformity and slit weir location on the generation of scour holes upstream of a slit weir. Bed topographical changes with time were recorded using a mechanical point gauge along the working section, and the velocity distribution was measured by a 2D current meter along the laboratory flume.

The experimental results manifest that the most efficient sedimentation release is caused by excessive shear stress and the creation of vortices upstream of the slit weir, if the slit is located either at the center or side of the weir. The progression of the scour phenomena continued with time until obtaining equilibrium conditions. The physical model well presents the scour development process under steady flow conditions for the implemented flow rates of 125, 95, 62, 50 and 34 L/s with a slit weir at the center and at side locations throughout 40 tests. Moreover, unsteady flow conditions were tested with a flow rate of 125, 62 and 34 L/s and the slit weir at the center position throughout 4 tests. All of the experienced scenarios were carried out with uniform and non-uniform sediment sizes of $d_{50} = 0.24$ and 0.55. The scour volume was obtained and matched for various cases, and this study it can be concluded with the following:

- The flow rate had a major impact on the resulting scour volume. Thus, when the flow rate increased from 34 L/s to 125 L/s, the scour volume increased 4 times for a

uniform sand of size $d_{50}$ = 0.24 mm at the center slit weir, and it was 3.25 when the slit was located at the side. For non-uniform sediment, the increment was 4 times for the same sand size at the center slit weir and three times at the side slit weir. The value of the increment in scour volume for uniform sediment with a median size of $d_{50}$ = 0.55 mm was four times at the center slit weir. In addition, the scour volume became 4.3 times larger than the value predicted with the minimum flow rate with the side slit weir. In addition, the difference for non-uniform sediment with a sand size of $d_{50}$ = 0.55 mm was 4 times at the center slit weir and 4.5 when the slit was at side of the weir.

- The effect of the median particle size played an essential role in the scour volume upstream of the slit weir. However, the scour volume was recorded as 2 times higher when adopting a sediment particle size of 0.24 mm compared to the values measured with a sediment size of 0.55 mm for uniform sediment at the center and side slit weirs. In addition, the difference was 3 times for non-uniform sediment when the slit was at the center of the weir and by 2 at the side slit weir.
- The influence of sand uniformity was investigated in this research for the same sand median size. The scour volume resulted in a higher value with uniform sediment compared to the value obtained with non-uniform ones by 25% when the sediment size was 0.24 mm and 30% with $d_{50}$ = 0.55 mm.
- The experimental work shows that the slit location had a governing impact on the scour volume. Higher values were recorded when the slit was positioned at the center of the weir, and the observed increment was 1.25 in the measured scour volume at the center slit weir compared to the values obtained when the slit was located at the side of the weir under the same conditions.

Further studies are recommended concerning the interactions among the maximum scour depth, scour volume and the turbulent kinetic energy for the efficient implementation of this study.

**Author Contributions:** Conceptualization, R.K.H.; methodology, R.K.H.; software, R.K.H.; validation, R.K.H.; formal analysis, R.K.H.; investigation, R.K.H.; resources, R.K.H.; data curation, R.K.H.; writing—original draft preparation, R.K.H.; writing—review and editing, R.K.H., A.A.-A. and T.A.M.; visualization, R.K.H.; supervision, A.A.-A. and T.A.M. All authors have read and agreed to the published version of the manuscript.

**Funding:** This research received no external funding.

**Data Availability Statement:** All data were collected from the experiments and field work presented in the manuscript.

**Conflicts of Interest:** The authors declare no conflict of interest.

**Abbreviations**

| | |
|---|---|
| $d_{50}$ | Median particle size |
| $d_s$ | Scour depth |
| $V_s$ | Scour volume |
| $h_s$ | Slit weir height |
| $h_w$ | Weir height |
| g | Gravity acceleration |
| $\mu$ | Dynamic viscosity |
| $\rho$ | Water density |
| $\rho_s$ | Sediment density |
| B | Flume width |
| $b_s$ | Slit weir width |
| v | Flow velocity |
| $v_a$ | Armor velocity |

v_c      Sediment entrainment critical velocity

Q      Flow rate

y      Flow water depth

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
