# Peer review of "Physical Modeling of the Scour Volume Upstream of a Slit Weir Using Uniform and Non-Uniform Mobile Beds"

_water, doi:10.3390/w14203273_

Round 1

Reviewer 1 Report

The paper is not suitable for publication because:

1-Through the paper, the subscriptions of variables have not been corrected written, for instance variables of Eq.(1).

2-Why did not Reynolds number play role in the estimation of the local scour depth? This issue should be clarified. 

3-Figure 1 should be reillustrated for better understanding

4-Literature review can be updated by :

-Effects of different geometric parameters of complex bridge piers on maximum scour depth: experimental study

-Experimental study of local scour around a vertical pier in cohesive soils

5-Authors should compare the present results with literature review in terms of quantitative and qualitative ways.

6-Table 3 requires the unit of measures.

7-How did authors compute equiblirium time of scour hole? 

8-Dimensional analysis needs various justifications such as representation of relevant references.

Reviewer 2 Report

This work develops Physical Modeling of Scour Volume Upstream of a Slit weir 2 Using Uniform and Non-uniform Mobile Beds. The authors provide new laboratory data to predict and simulate the time-varying scour upstream slit weir for efficient sediment release from hydropower intake. This work then systematically analyze the effect of uniform and non-uniform sediment with varying sediment size. This work is interesting for the audience in Water and also performed a very comprehensive literature review. Some comments below may help the author to further improve this manuscript.

  In line 280, The author mention `horseshoe vortices'. How could the author see the horseshoe vortices from the visualization?   Some notation is not used consistently. For example, d50 and $d_{50}$ are used alternatively. It is suggested to use the subscript consistently throughout the paper.   It is suggested to explicitly present the definition of a non-dimensional parameter such as the Froude number.   This paper also needs some language polishing. The third line of the abstract, `temporal and special'->'temporal and spatial'?   line 19, `It demonstrate'->`It demonstrates'   Table 1, in the row with reference `Marion et al. (2006)', something is happening there that hides the text.   Figures 17 and 18, top row, label C is also partially hidden.

Round 2

Reviewer 1 Report

Accept as is